# TF-JEPA: Predictive Alignment of Time–Frequency Representations Without Contrastive Pairs

## Abstract

Learning generalizable representations from multivariate time series is challenging due to complex temporal dynamics, distribution shifts, and the difficulty of effectively designing contrastive pairs. We introduce TF-JEPA, a noncontrastive self-supervised method that leverages predictive alignment to integrate representations from the time and frequency domains without relying on negative sampling. Specifically, TF-JEPA utilizes dual online encoders for time and frequency domains, each paired with its own momentum-updated target encoder, embedding both views into a stable and unified latent space. Unlike conventional contrastive methods, this predictive approach enables full end-to-end fine tuning for downstream adaptation. Experimental results on diverse real world datasets, including sleep EEG classification, gesture recognition, mechanical fault detection, and biosignal-based muscle response classification, demonstrate that TF-JEPA matches or surpasses contrastive and time frequency consistency baselines. TF-JEPA improves macro F1 scores by up to 8.6 percentage points while also reducing GPU memory consumption by approximately 35%. These findings illustrate the promise of predictive alignment as a broadly applicable and modality agnostic framework for self supervised learning beyond traditional contrastive methods.

## 1 Introduction

Learning effective representations from time-series data is a fundamental yet challenging problem in modern machine learning. Such data arise in critical domains, including healthcare, transportation, and finance but differ markedly from images or text. Temporal dependencies, non-stationarity, and frequent domain shifts across datasets hinder generalization Ismail Fawaz et al. (2018); Gupta et al. (2021). Moreover, labeled time-series are often scarce and costly to obtain, especially in medical settings that require expert annotation Harutyunyan et al. (2019). Transfer learning has emerged as a powerful paradigm in time-series modeling, enabling pre-trained representations to generalize across domains Ye & Dai (2021). Unlike vision or text, time-series signals possess a natural time-frequency duality that many representation learning methods have yet to fully exploit. This duality is particularly critical in physiological signals such as EEG Zhang & Yao (2021), where both spectral and temporal features are diagnostically relevant. Classical signal processing has long used time–frequency analysis to interpret non-stationary data Cohen (1995); Papandreou-Suppappola (2018), with FFT serving as the foundational transformation Brigham (1988). These ideas have inspired recent adaptations in neural time-series modeling Cheng et al. (2021). These factors motivate self-supervised learning approaches capable of leveraging abundant unlabeled data and facilitating transfer across tasks.

Contrastive learning has become the dominant self-supervised paradigm for time-series: it pulls together augmented views of the same sample (positive pairs) while pushing apart different samples (negative pairs) Chen et al. (2020); van den Oord et al. (2019). However, applying contrastive learning to time-series is particularly difficult because suitable augmentations and negative-pair selection are challenging to design Zhang et al. (2022); Wickstrøm et al. (2022). These methods are sensitive to augmentation choice, require large batch sizes or memory banks, and are often evaluated on a single dataset, limiting cross-domain transferability Chen et al. (2020).

Recent non-contrastive approaches, notably the Joint Embedding Predictive Architecture (JEPA) Le-Cun (2023), have shown that strong representations can be learned without explicit negative pairs. In one approach, JEPA trains an online network to predict a momentum-updated target network's representation of the same sample under different augmentations, sidestepping negative sampling and achieving state-of-the-art results in vision. Predictive objectives of this kind have not yet been systematically explored for timeseries data, where the natural dual view of time and frequency gives a compelling test bed. Bridging this gap calls for objectives that can integrate complementary views in any modality; time–frequency alignment therefore serves as an ideal task and the focus of this work.

A notable recent effort is Time–Frequency Consistency (TF-C) Zhang et al. (2022), which aligns time and frequency domain embeddings with a contrastive objective. TF-C showed that incorporating spectral structure can aid cross-domain generalization. At the same time, contrastive training introduces a dependence on cross-sample negatives (and thus large effective batch sizes or memory banks), sensitivity to augmentation and temperature choices, and the possibility of penalizing semantically similar "false negatives". Because fine-tuning protocols vary in the literature, we report both linear-probe and full end-to-end fine-tuning results for TF-C in our comparisons.

In this work, we introduce TF-JEPA (Time-Frequency Joint Embedding Predictive Architecture), a non-contrastive self-supervised framework that aligns time and frequency representations through prediction rather than contrastive repulsion. First, we introduce a momentum-based dual-encoder architecture, consisting of an online time encoder and a momentum-updated frequency encoder. The momentum encoder provides stable predictive targets through exponential moving average updates. Second, predictive alignment eliminates negative pairs, thereby avoiding instance discrimination pitfalls. Finally, because TF-JEPA avoids contrastive collapse, the entire model remains trainable during downstream fine-tuning, allowing full adaptation to the target data distribution.

We evaluate TF-JEPA on diverse real-world benchmarks, including sleep EEG with epilepsy, fault detection, and gesture-recognition datasets. Our experiments show consistent improvements over self-supervised methods such as TF-C, improving accuracy and F1 significantly on some datasets. These results highlight the advantages of non-contrastive predictive objectives for robust time-frequency alignment.

In summary, our contributions are threefold: (1) we propose a momentum-based dual-encoder architecture for time-series that aligns time and frequency domain representations without negative pairs, (2) we demonstrate that this predictive alignment strategy yields transferable embeddings suitable for end-to-end fine-tuning, and (3) we achieve competitive or superior performance compared to existing methods on multiple real-world time-series benchmarks.

## 2 FROM TF-C TO TF-JEPA

Time–frequency consistency (TF–C) established that aligning a waveform with its own spectrum can improve cross-dataset transfer in biosignal analysis. Yet TF–C depends on a contrastive objective whose computational and methodological demands have become increasingly restrictive. Contrastive learning requires large batches or memory queues, stores an $\mathcal{O}(B^2)$ similarity matrix, and, in practice, is vulnerable to "false negatives" in which two nearly identical signals are pushed apart.

Subsequent frequency-aware variants reduce some of these drawbacks but introduce bespoke components. Examples include masked frequency auto-encoders Liu et al. (2024) and learnable Fourier filters, which rely on task-specific masking schemes that limit reuse.

TF–JEPA replaces the contrastive repulsion paradigm with predictive alignment, built on three design choices:

1. **Dual EMA targets.** A frozen time encoder and a frozen frequency encoder are updated after every step by an exponential moving average (EMA, momentum $m = 0.995$) of the online weights, providing stable target representations with no gradient overhead.

2. **Lightweight predictors.** Two small multilayer perceptrons, each mapping $\mathbb{R}^{128} \to \mathbb{R}^{128}$, transform the online embeddings so that they predict the corresponding target view. A

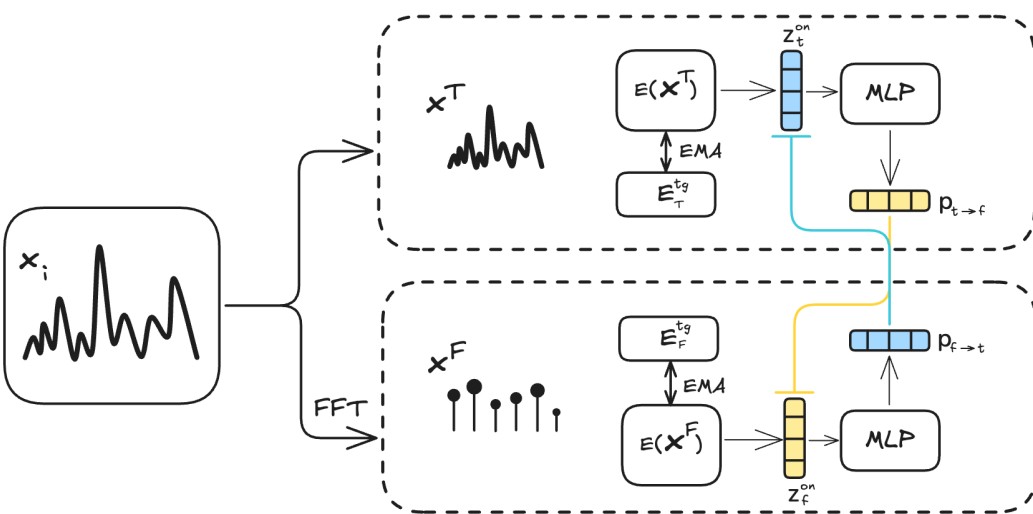

Figure 1: Architecture diagram for pre-training steps of TF-JEPA. This diagram communicates the three key ideas (i) time/frequency dual encoders, (ii) EMA targets, (iii) two cross-view predictors.

BYOL-style Grill et al. (2020) cosine loss

$$\mathcal{L} = \sum_{v \in \{t,f\}} \left\| p_{v \to \bar{v}} - z_{\bar{v}}^{\text{tg}} \right\|_{\cos}$$

aligns the two domains without negative pairs or large batch queues.

3. **End-to-end fine-tuning.** Because the objective avoids contrastive collapse, all encoder weights can be unfrozen during downstream training, allowing full adaptation to the target distribution (for example, SleepEEG $\to$ Epilepsy or HAR $\to$ Gesture).

TF–JEPA retains TF–C's intuition of cross-view alignment while reducing GPU memory by approximately 35% on 178-step EEG windows, operating with batches as small as 32, and improving cross-dataset transfer macro-$F_1$ by up to eight percentage points (for example, Fault Detection A $\to$ B).

## 2.1 WHY PREDICTIVE ALIGNMENT? INTUITION BEHIND TF–JEPA

**Time and frequency as complementary "modalities".** A discrete time–series and its Fourier spectrum form two loss-less, invertible views of the same signal. Similar to image–text pairs in CLIP Radford et al. (2021) or audio–visual pairs in AVID Arandjelović & Zisserman (2017), these dual views emphasize different statistical regularities: the time domain exposes local temporal dynamics (e.g., waveform shape, transients), whereas the frequency domain highlights global rhythmic structure and stationarity. Leveraging both views therefore offers a built-in multi-modal supervision signal without requiring paired datasets from different sensors.

**From contrastive repulsion to cross-view prediction.** Contrastive objectives enforce invariance by repelling all other samples in the mini-batch, which costs $\mathcal{O}(B^2)$ memory and can mistreat near-duplicates as negatives. Joint-Embedding Predictive Architectures (JEPA) LeCun (2023) invert that idea: each online encoder predicts the latent vector produced by a slow-moving EMA target encoder of the opposite view. Concretely, the time encoder $E_t^{\text{on}}$ learns to match the frequency target $z_f^{\text{tg}} = E_f^{\text{tg}}(x_f)$, while the frequency encoder $E_f^{\text{on}}$ predicts the time target $z_t^{\text{tg}} = E_t^{\text{tg}}(x_t)$. This removes the need for negatives, keeps memory linear in $B$, and, like BYOL Grill et al. (2020), prevents collapse because the EMA targets evolve slowly yet non-trivially. Applying JEPA across time/frequency views yields three benefits

Table 1: Target-task performance (%). **NormWear** and **CBraMod** are foundation models providing large-scale pre-trained physiological priors. NormWear is pre-trained on diverse wearable modalities; CBraMod is pre-trained on a large EEG corpus using criss-cross spatial–temporal attention and masked patch reconstruction. Both are fine-tuned only on each target dataset under identical heads and optimizers, without source→target transfer, serving as adaptation context for TF-JEPA.

| Dataset | NormWear | | | | CBraMod | | | |
| --- | --- | --- | --- | --- | --- | --- | --- | --- |
| | AUC | AP | Acc. | F1 | AUC | AP | Acc. | F1 |
| Epilepsy | 98.21 | 99.42 | 95.51 | 92.61 | 98.02 | 99.10 | 90.35 | 97.23 |
| FD-B | 84.54 | 67.15 | 58.30 | 61.56 | 71.14 | 64.75 | 75.49 | 65.58 |
| Gesture | 88.56 | 64.33 | 55.00 | 49.04 | 92.34 | 77.89 | 74.17 | 73.56 |
| EMG | 93.73 | 83.85 | 87.71 | 62.39 | 99.83 | 99.46 | 98.04 | 97.64 |

1. **Semantic alignment.** Predicting one view from the other forces the network to focus on view-invariant factors (sleep stage, bearing damage, gesture identity) while disregarding nuisance details specific to either domain.

2. **Stability without collapse.** EMA targets provide a non-trivial prediction signal that evolves slowly; empirical and theoretical analyses Tian et al. (2021); Bardes et al. (2022) show this circumvents trivial-solution collapse even with small batches.

3. **Linear complexity.** No $B \times B$ similarity matrix or memory queue is formed, so memory and compute scale linearly with $B$.

**Why alignment should emerge self-supervised.**    Because the FFT is invertible, all task-relevant information in one view is present in the other. Minimizing the cosine distance between predicted and target embeddings therefore bounds the mutual information between the views from below Poole et al. (2019); the optimum is reached when each encoder concentrates that shared information into its latent code. In practice we observe that the resulting representations cluster by semantics across datasets, echoing the theoretical expectation that view agreement acts as an information bottleneck selecting factors that generalize across domains.

**Relation to prior multi-modal JEPA work.**    Concurrent studies have applied predictive objectives to RGB–depth pairs Assran et al. (2022) and image–audio pairs Alayrac et al. (2022). TF–JEPA is the first to exploit the intrinsic duality of a single signal, requiring no additional sensors or annotators. This property makes the method attractive for domains (e.g. medical telemetry, vibration monitoring) where extra modalities are costly or infeasible to collect.

## 3    PROPOSED METHOD

As summarized previously, TF–JEPA learns a shared representation for raw time-series and their spectra without relying on negative pairs. Two encoders: one operating in the time domain and one in the frequency domain—are trained so that each predicts the other's output through momentum-updated target networks, providing stable signals during optimization.

### 3.1    MODEL

**Encoders.**    For every sample we form two views: a time-domain sequence $x_t \in \mathbb{R}^{B \times T \times C}$ and its frequency-domain counterpart $x_f = |\text{FFT}(x_t)|$. Following the TF-C implementation[1], we compute a magnitude-only spectrum over the full segment (no STFT), with FFT size N equal to the sequence length defined in Appendix A. The phase information is discarded, and spectra are not normalized across the training set. During pre-training, frequency augmentations randomly zero out or add noise to 10% of frequency bins, while time-domain augmentations apply jittering with $\sigma = 0.8$. Each view

---

[1]We note that while the TF-C paper describes using targeted single-component perturbations (E=1) with conditional boosting ($\alpha = 0.5$), their publicly available implementation uses a simpler approach that we adopt here for fair comparison.

Table 2: Transfer performance (%). **TS-TCC**[*], **TF-C**, and **TF-JEPA**[†] are pre-trained only on the single source dataset indicated (column 1) and then fine-tuned on the corresponding target dataset, following identical transfer-learning protocols. This setup allows direct comparison among three models of similar size, each using substantially less pre-training data than the foundation models in Table 1. The right-most column reports the margin of TF-JEPA over the best competing transfer baseline on each task; positive values favor TF-JEPA.

| Transfer task | TS-TCC | | | | TF-C | | | | TF-JEPA[†] | | | | $\Delta$F1 | $\Delta$Acc |
|---|---|---|---|---|---|---|---|---|---|---|---|---|---|---|
| | AUC | AP | Acc. | F1 | AUC | AP | Acc. | F1 | AUC | AP | Acc. | F1 | | |
| SleepEEG→Epilepsy | 96.27 | 86.23 | 85.88 | 82.48 | 98.11 | **94.56** | 94.95 | 91.49 | **99.07** | 94.51 | **95.31** | **92.24** | ↑0.75 | ↑0.36 |
| FD-A→FD-B | 85.23 | 83.80 | 73.85 | 77.31 | 94.35 | 92.09 | 89.34 | 91.62 | **99.98** | **99.47** | **99.28** | **99.47** | ↑7.85 | ↑9.94 |
| HAR→Gesture | 86.60 | 65.61 | 63.33 | 59.91 | 89.55 | 65.91 | 68.33 | 65.79 | **91.47** | **73.16** | **75.66** | **74.34** | ↑8.55 | ↑7.33 |
| ECG→EMG | **96.35** | **85.19** | 85.88 | **82.48** | 87.53 | 82.74 | 85.37 | 80.51 | 92.53 | 79.41 | **87.80** | 80.03 | ↓2.45 | ↑1.92 |

is processed by an identical $L$-layer one-dimensional Transformer encoder with model dimension $d_{\text{model}}$. After the Transformer, mean pooling over the temporal axis followed by a two-layer MLP projector produces latent vectors

$$z_t^{\text{on}}, \; z_f^{\text{on}} \in \mathbb{R}^{d_z}, \qquad d_z = 128.$$

**Momentum targets.** Frozen target encoders $G_t^{\text{tg}}$ (time) and $G_f^{\text{tg}}$ (frequency) are updated after every optimization step by an exponential moving average (EMA) of the online encoder weights:

$$\theta^{\text{tg}} \leftarrow m\,\theta^{\text{tg}} + (1-m)\,\theta^{\text{on}}, \qquad 0.995 \le m \le 0.9995.$$

Because these target encoders are never back-propagated through, they add minimal memory and no optimizer state while outputting the reference embeddings $z_t^{\text{tg}}$ and $z_f^{\text{tg}}$.

**Predictors.** Two lightweight predictor MLPs with dimensions $128 \to 256 \to 128$ are applied to the online embeddings. The time-view code is mapped to $p_{t\to f} = P_{t\to f}(z_t^{\text{on}})$ and trained to match the target frequency embedding $z_f^{\text{tg}}$. Symmetrically, the frequency-view code is mapped to $p_{f\to t} = P_{f\to t}(z_f^{\text{on}})$ and trained to match $z_t^{\text{tg}}$. Introducing such predictors, as in BYOL, helps stabilize training and prevents representational collapse.

### 3.2 Loss

The objective is the sum of two BYOL-style cosine similarity terms,

$$\mathcal{L}_{\text{TF-JEPA}} = \mathcal{L}_{\cos}(p_{t\to f}, z_f^{\text{tg}}) + \mathcal{L}_{\cos}(p_{f\to t}, z_t^{\text{tg}})$$

where,

$$\mathcal{L}_{\cos}(p, z) = 2 - 2 \cdot \frac{p \cdot z}{\|p\|_2 \|z\|_2}$$

for each directional prediction. Maximizing cosine similarity aligns the two domains without requiring negative samples.

## 4 Experiments and Results

### 4.1 Experimental setup

We evaluate TF-JEPA on four widely-used cross-dataset transfer tasks in time-series representation learning. Each non-foundational model (TF-JEPA, TF-C, and TS-TCC Eldele et al. (2021)) is pre-trained exclusively on the specified source dataset using the recommended hyperparameters from their respective papers, and then fine-tuned on the corresponding target dataset with identical classifier heads. To ensure direct comparability, the classifier architecture, latent dimension $d_z = 128$, and optimizer hyperparameters remain consistent across methods during fine-tuning. TF-JEPA employs a smaller batch size of 32 due to its predictive alignment approach, while TF-C and TS-TCC require a larger batch size of 128 to sufficiently sample negative pairs during contrastive training. This

assertion is confirmed with an ablation study across 6 batch sizes from 16 to 512. For example, with HAR transfer experiment TF-JEPA demonstrates robust performance across all batch sizes with a coefficient of variation of 2.05%, and accuracy saturating at around 76% for batch sizes $\geq 64$. Our choice of batch size 32 achieves competitive performance (75.66% accuracy, 91.47% AUC) while requiring significantly less memory than contrastive methods, with only a 0.35 percentage point accuracy trade-off compared to the saturation point. All experiments were conducted on a single NVIDIA A10 GPU (32 GB memory) using mixed-precision training.

We also select NormWear Luo et al. (2024) as our baseline state-of-the-art foundation model specifically tailored for wearable-sensing data, capable of extracting generalized, modality-agnostic representations from a diverse array of physiological signals (PPG, ECG, EEG, GSR, IMU). Its broad pre-training across multiple physiological signals and demonstrated effectiveness in various transfer scenarios provides a strong benchmark for evaluating generalizable representations.

We further include CBraMod Wang et al. (2025), a newly introduced brain foundation model for EEG decoding. Similar to NormWear, CBraMod is first pre-trained on a large heterogeneous corpus, here exclusively EEG, using a criss-cross Transformer backbone with parallel spatial–temporal attention and conditional masked EEG reconstruction on patch tokens. Following the same downstream adaptation role as NormWear, CBraMod is then fine-tuned solely on each target dataset (no source→ target transfer fine-tuning is performed). This provides a second foundation reference point that measures how well generalized priors from large-scale EEG pre-training can adapt to downstream decoding tasks under the same target fine-tuning protocol used for NormWear, allowing TF-JEPA to be contextualized against both broad wearable pre-training and large EEG-only pre-training priors.

TS-TCC, another contrastive learning method, was chosen due to its methodological similarity to TF-C and popularity as a representation-learning approach that explicitly addresses temporal dynamics and contextual relationships within time-series data. TF-C, our primary contrastive baseline, directly motivates TF-JEPA. It emphasizes time-frequency consistency, aiming to embed time-based and frequency-based representations of an example closely together within a shared latent space through contrastive methods. Evaluating against TF-C allows us to explicitly measure the impact and advantages of our proposed non-contrastive predictive alignment approach.

Together, these four methods, NormWear and CBraMod (generalized foundation models), TS-TCC (temporal-contextual contrastive), and TF-C (time-frequency consistency), provide a comprehensive benchmark spectrum. This range ensures a thorough evaluation of TF-JEPA's ability to achieve robust and generalizable representations without reliance on contrastive pairs, highlighting both methodological innovation and practical advantages in computational efficiency and downstream performance.

## 4.2 TRANSFER LEARNING PERFORMANCE

Table 2 reports accuracy and macro-$F_1$ on the target datasets.

1. **SleepEEG→Epilepsy.** Transfer from 82 healthy overnight EEG recordings to seizure detection in 500 subjects—a shift from benign to pathological patterns.

2. **FD-A→FD-B.** Bearing-fault detection across two operating regimes with different torque and speed, testing robustness to mechanical covariate shift.

3. **HAR→Gesture.** Daily full-body motions (50 Hz, nine channels) to fine-grained hand gestures ($\approx 100$ Hz, three channels), probing scale and granularity gaps.

4. **ECG→EMG.** Cross-organ physiological transfer: single-lead cardiac rhythms (300 Hz) to tibialis-anterior electromyograms (4 kHz).

TF–JEPA surpasses contrastive methods on SleepEEG→Epilepsy and on both domains of the Fault Detection benchmark and Gesture recognition, improving macro-$F_1$ by more than eight percentage points. TF-JEPA falls slightly short in the cross-organ physiological transfer task and a deeper analysis is shown below.

As shown in Figure 2, we notice that performance improves with higher EMA momentum $m$: we observe a positive correlation between $m$ and transfer metrics (Pearson $r = 0.833$ across settings), with all metrics peaking at $m = 0.9995$. With 3 seeds for each $m$ and a 95% CI on $\Delta$F1, the best

Table 3: Dataset statistics. $C$ = number of classes after any relabelling; $S$ = sampling rate; $N_{\text{pre}}$ / $N_{\text{ft}}$ give pre-training and fine-tuning sample counts. Window lengths follow cited preprocessing protocols.

| Dataset | Domain | $C$ | $S$ (Hz) | Window | $N_{\text{pre}}$ | $N_{\text{ft}}$ |
|---------|--------|-----|----------|--------|------------------|-----------------|
| SleepEEG | EEG (sleep) | 5 | 100 | 200 | 371 055 | – |
| Epilepsy | EEG (seizure / normal) | 2 | 178 | 178 | – | 60 |
| FD-A | Vibro-acoustic (cond. A) | 3 | 64 k | 5 120 | 18 882 | – |
| FD-B | Vibro-acoustic (cond. B) | 3 | 64 k | 5 120 | – | 18 864 |
| HAR | 9-axis IMU (daily activity) | 6 | 50 | 128 | 10 299 | – |
| Gesture | 3-axis accel. (hand motion) | 8 | $\sim$100 | 256 | – | 440 |
| ECG | Cardiac rhythm | 4 | 300 | 1 500 | 8 528 | – |
| EMG | Tibialis-anterior EMG | 3 | 4 000 | 1 500 | – | 163 |

setting ($m = 0.9995$) exceeds the worst by $+11.3$pp in the HAR transfer experiment. This pattern generalizes across datasets: ECG shows the most dramatic sensitivity with a 39 percentage point improvement ($53.7\% \rightarrow 92.7\%$ accuracy), while SleepEEG exhibits optimal performance at the slightly lower $m = 0.995$ (90.8% accuracy). The dataset-dependent optimal momentum suggests that signal complexity influences the required target network stability. Biomedical time series with intricate temporal patterns (ECG, HAR) benefit most from ultra-slow updates ($m = 0.9995$), while sleep data achieves peak performance with moderate stability ($m = 0.995$). Intuitively, ultra-slow target updates stabilize the non-contrastive objective, improving stability and the signal-to-noise ratio in the target representations. The consistent superiority of high momentum values ($m \geq 0.995$) across all datasets validates the critical importance of target network stability in BYOL-style self-supervised learning for time series, with the EMA update rate of 0.05% or less proving optimal for complex temporal patterns.

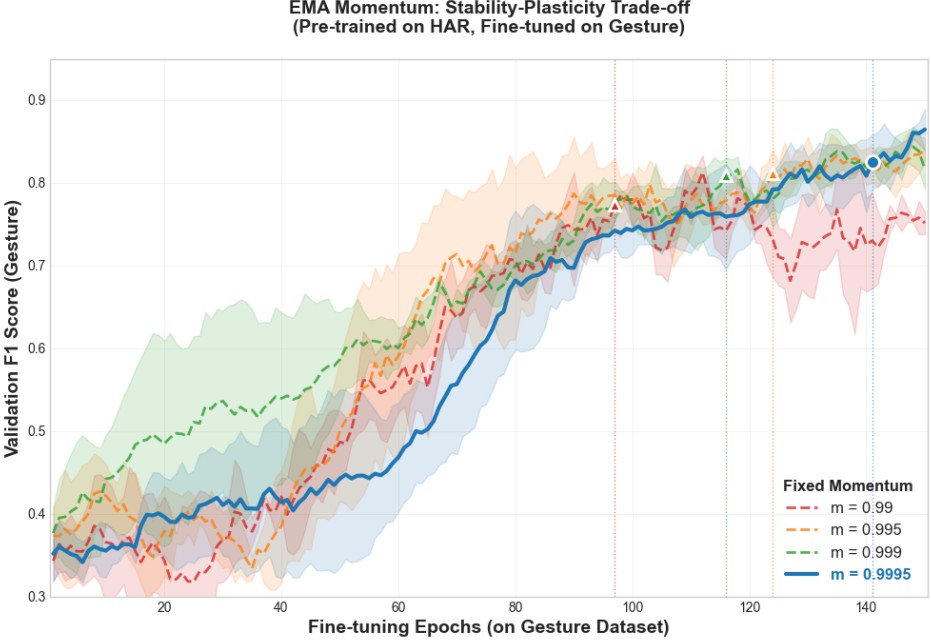

Figure 2: Validation F1 on Gesture (fine-tuning) after pre-training on HAR with fixed EMA momenta. Dotted lines show with 3 seeds for each $m$ and 95% arrival epochs, $m = 0.9995$ converges more slowly than lower $m$ but yields the highest final score, so we adopt it when final accuracy is prioritized over time-to-stability.

We evaluate TF-JEPA on four diverse transfer-learning scenarios, each highlighting distinct challenges in generalization across physiological and mechanical domains. The SleepEEG to Epilepsy task tests transfer from structured, healthy sleep EEG patterns to pathological seizure detection. FD-A to FD-B examines robustness in industrial fault diagnostics across different mechanical operating conditions Lessmeier et al. (2016). The ECG to EMG transfer explores physiological cross-modality generalization from cardiac rhythms to muscle activation signals, despite significant organ-specific variations Clifford et al. (2017). Lastly, the HAR to Gesture task evaluates whether generalized motion features learned from daily human activities can facilitate recognition of fine-grained symbolic hand gestures Anguita et al. (2013). Collectively, these tasks comprehensively test TF-JEPA's ability to extract representations that generalize across modalities, physiological states, and operational conditions.

### 4.3 ANALYSIS OF THE ECG TRANSFER CASE

The ECG→EMG transfer has three classes labeled 0, 1, and 2. As shown in Figure 3, TF-JEPA identifies class 2 reliably but frequently predicts label 1 when the ground truth is 0, leading to the observed macro-$F_1$ drop. Classes 0 and 1 differ mainly by subtle waveform-shape variations; the explicit repulsion term in TF-C appears to preserve this fine boundary, whereas TF-JEPA's predictive loss focuses on cross-view alignment and is less sensitive to inter-sample separation. Introducing a class-balanced sampling during fine-tuning may help recover this distinction, and we leave that exploration to future work.

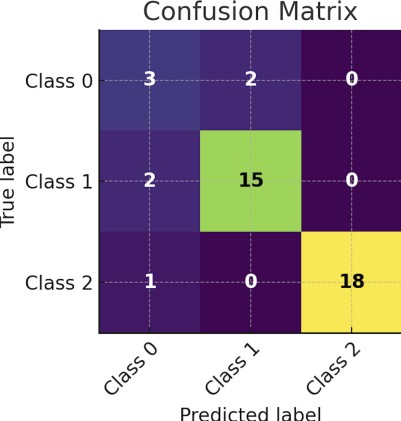

Figure 3: Confusion matrix for the 3-class test set (41 samples). Diagonal cells give correct predictions: class 0: 3/5, class 1: 15/17, class 2: 18/19 while off-diagonal counts expose the main failure mode. Class 0 & class 1 confusions (2 + 2 cases). Color intensity scales with sample count for quick visual emphasis.

### 4.4 RESOURCE USAGE

Because TF–JEPA eliminates the quadratic $B \times B$ similarity / logit tensor required by the NT–Xent loss, it trains 178-step EEG windows with a batch size of 32 in 3.4 GB of GPU memory, versus 5.3 GB for TF–C, and delivers a 1.6× speed-up on an NVIDIA A10G. When the batch size is held constant, removing that tensor still lowers peak memory by roughly 10–15 % and yields a 1.2–1.4× throughput gain. Note that TF–C keeps its negatives entirely within the current mini-batch, so the only memory reclaimed is the pair-wise logits; no separate negative queue is involved.

These efficiency gains come without sacrificing accuracy: TF-JEPA matches or outperforms TF-C on two of four challenging cross-dataset transfers and stays competitive on the others, underscoring predictive alignment as a lean, modality-agnostic alternative to contrastive objectives for self-supervised learning on structured time-series data.

## 5 CONCLUSION

This work introduces TF-JEPA, a predictive, non-contrastive framework for learning shared time–frequency representations from unlabeled time-series data. By coupling an online time encoder with a momentum-updated frequency encoder and training them with a lightweight cosine loss, TF-JEPA removes the need for negative pairs, lowers GPU memory by up to thirty-five percent, and improves cross-dataset transfer performance by as much as eight percentage points on representative benchmarks. Because the objective is stable without a contrastive repulsion term, all encoder weights remain trainable during downstream fine-tuning, enabling full adaptation to target distributions.

Future directions include scaling the method to longer sequences and additional modalities, integrating predictive alignment with complementary masked-reconstruction objectives, and analyzing the few tasks where TF-JEPA underperforms contrastive baselines in order to further strengthen its generality.

### ETHICS STATEMENT

This work uses only publicly available, previously released datasets as cited in the paper; to the best of our knowledge these datasets are de-identified and were collected under the original providers' approvals and terms of use. We did not collect new human-subject data, perform interventions, or attempt re-identification. Potential risks include misuse of models for clinical or safety-critical decisions; our models are research prototypes and are not intended for real-time medical, industrial, or safety-critical deployment without appropriate validation. We report results fairly, include negative/neutral findings where applicable (e.g., transfer tasks where performance lags), and disclose settings that materially affect results (e.g., batch size, momentum). We follow dataset licenses/terms and respect privacy. We are not aware of conflicts of interest or external sponsorship that could bias the work. Fairness concerns may arise from dataset shift and class imbalance; we partially address these via cross-dataset evaluation and ablations, and we encourage further audits with demographically annotated datasets.

### REPRODUCIBILITY STATEMENT

We provide all training and evaluation details needed to reproduce results. Architectures, data processing, and loss are specified in Sections 3–4; full hyper-parameters and training schedules for TF-JEPA, TF-C, TS-TCC, and NormWear are listed in Tables 4, 7, 8, and 5. Dataset choices, window lengths, and class counts appear in Table 3. We report hardware and software versions in the appendix (Appendix A), and we fix random seeds. An anonymized code archive (training scripts, configs, and evaluation) is included in the supplementary material to facilitate end-to-end replication of the reported experiments.

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

## A  APPENDIX A. TF-JEPA EXPERIMENTAL SETTINGS

Unless stated otherwise, all experiments were run on a single NVIDIA A10G-32 GB GPU using PyTorch 2.7.0 + CUDA 12.8. Reproducibility is ensured by fixing the random seed to 42.

Table 4: Key hyper-parameters for **TF-JEPA**. $d_{\text{model}}$ is the Transformer embedding dimension (equal to the aligned sequence length). "Batch / LR" list values for self-supervised pre-training (P) and supervised fine-tuning (F). All runs use dropout $= 0.35$.

| Experiment (P → F) | $d_{\text{model}}$ | Channels | Momentum $m$ | Batch (P/F) | LR (P/F) | Epochs (P/F) |
|---|---|---|---|---|---|---|
| SleepEEG → Epilepsy | 178 | 1 | 0.995 | 128 / 60 | $3 \times 10^{-4}$ / $3 \times 10^{-4}$ | 10 / 100 |
| FD-A → FD-B | 5120 | 1 | 0.9995 | 64 / 60 | $3 \times 10^{-4}$ / $3 \times 10^{-4}$ | 10 / 100 |
| HAR → Gesture | 206 | 1 | 0.9995 | 128 / 42 | $3 \times 10^{-4}$ / $3 \times 10^{-4}$ | 10 / 100 |
| ECG → EMG | 1500 | 1 | 0.9995 | 128 / 41 | $3 \times 10^{-6}$ / $3 \times 10^{-6}$ | 10 / 100 |

Table 5: Hyper-parameters for **NormWear** fine-tuning. All runs use masking ratio $= 0.8$, patch size $(9, 5)$, dropout $= 0.35$.

| Target Dataset | Seq. Len. | Channels | Batch | LR | Epochs |
|---|---|---|---|---|---|
| Epilepsy | 178 | 1 | 16 | $1 \times 10^{-2}$ | 100 |
| FD-B | 21 | 1 | 8 | $1 \times 10^{-3}$ | 100 |
| Gesture | 315 | 3 | 32 | $1 \times 10^{-3}$ | 100 |
| EMG | 96 | 1 | 32 | $1 \times 10^{-3}$ | 100 |

Table 6: Hyper-parameters for **CBraMod** fine-tuning. All runs use dropout $= 0.1$, weight decay $= 5 \times 10^{-2}$, pretrained weights.

| Target Dataset | Seq. Len. | Channels | Batch | LR | Epochs |
|---|---|---|---|---|---|
| Epilepsy | 178 | 1 | 8 | $1 \times 10^{-4}$ | 50 |
| FD-B | 5120 | 1 | 8 | $1 \times 10^{-4}$ | 50 |
| Gesture | 206 | 3 | 8 | $1 \times 10^{-4}$ | 50 |
| EMG | 1500 | 1 | 8 | $1 \times 10^{-4}$ | 50 |

Table 7: Hyper-parameters for **TFC**. Temperature $= 0.2$, dropout $= 0.35$.

| Experiment (P → F) | Seq. Len. | Channels | Batch (P/F) | LR (P/F) | Epochs (P/F) |
|---|---|---|---|---|---|
| SleepEEG → Epilepsy | 178 | 1 | 128 / 60 | $3 \times 10^{-4}$ / $3 \times 10^{-4}$ | 10 / 100 |
| FD-A → FD-B | 5120 | 1 | 64 / 60 | $3 \times 10^{-4}$ / $3 \times 10^{-4}$ | 10 / 100 |
| HAR → Gesture | 206 | 1 | 128 / 42 | $3 \times 10^{-4}$ / $3 \times 10^{-4}$ | 40 / 100 |
| ECG → EMG | 1500 | 1 | 128 / 41 | $3 \times 10^{-6}$ / $3 \times 10^{-6}$ | 100 / 100 |

Table 8: Hyper-parameters for **TS-TCC**. Temperature $= 0.2$, dropout $= 0.35$.

| Experiment (P → F) | Win. Len. | Channels | Batch (P/F) | LR (P/F) | Epochs (P/F) |
|---|---|---|---|---|---|
| SleepEEG → Epilepsy | 178 | 1 | 32 / 16 | $3 \times 10^{-4}$ / $3 \times 10^{-4}$ | – / 80 |
| FD-A → FD-B | 5120 | 1 | 64 / 16 | $3 \times 10^{-4}$ / $3 \times 10^{-4}$ | 40 / 40 |
| HAR → Gesture | 206 | 3 | 64 / 64 | $3 \times 10^{-7}$ / $3 \times 10^{-7}$ | 5 / 5 |
| ECG → EMG | 1500 | 1 | 32 / 16 | $3 \times 10^{-6}$ / $3 \times 10^{-4}$ | 10 / 20 |

