# OpenReview forum: "TF-JEPA: Predictive Alignment of Time–Frequency Representations Without Contrastive Pairs"
_ICLR.cc/2026/Conference — Submitted to ICLR 2026_

### Official Review · Reviewer_pfDZ · 2025-10-28

**Soundness:** 3
**Presentation:** 3
**Contribution:** 2
**Rating:** 4
**Confidence:** 4

**Summary:**

The paper introduces TF-JEPA, a non-contrastive method for learning shared time-frequency representations from unlabeled time series. It reduces GPU memory, and sometimes improves cross-dataset transfer.

**Strengths:**

- Non-contrastive self-supervised method based on time-frequency representations
- Well-written simple paper
- Good results on cross-domain transfer

**Weaknesses:**

- Somewhat limited novelty as JEPA and predictive alignment is not new. The benchmarks are also not novel as they're borrowed from TF-C paper.
- Since results are mixed, the paper would benefit from studies on more datasets.
- As [3] shows, TF-C can be used for additional downstream tasks, such as clustering and anomaly detection. TF JEPA has not been evaluated in these tasks, so generalizability of the learned representations is limited.
- The authors claim in the abstract that the method is "broadly applicable", but no findings on pretraining and fine tuning has been demonstrated, see for example recent works [1],[2]. Given the growing literature in time series foundation models, it would be beneficial to compare against such models, and even examine pretraining.

[1] S. Gao et al, "UniTS: A Unified Multi-Task Time Series Model", NeurIPS 2024

[2] M. Goswami et al, "MOMENT: A Family of Open Time-series Foundation Models", ICML 2024

[3] X. Zhang et al, "Self-Supervised Contrastive Pre-Training for Time Series via Time-Frequency Consistency", NeurIPS 2022

**Questions:**

- Why does TF-JEPA underperform on some datasets? Where is it expected to do better and why?
- Would TF-JEPA be able to be used as a pretraining objective on heterogeneous time series, similar to UniTS/MOMENT?
- How does TF-JEPA do on clustering and anomaly detection tasks?

---

> ### Author Response · Authors · 2025-12-03
>
> Thanks for the thoughtful feedback. A few clarifications.
>
> TF-JEPA is not intended as a brand-new architecture but as a demonstration that predictive alignment can replace contrastive repulsion in the time–frequency setting while remaining competitive across cross-dataset transfers and enabling full end-to-end fine-tuning with substantially lower compute. The benchmarks follow TF-C intentionally so the comparison is controlled. We agree the novelty is incremental rather than architectural, and we will state this more explicitly.
>
> Regarding mixed results: TF-JEPA improves performance on most transfer tasks, and we analyze the one underperforming case in the paper (ECG→EMG). Predictive alignment emphasizes shared cross-view structure, which helps under distribution shift but can blur very fine inter-class distinctions; we will clarify where this tradeoff occurs.
>
> On broader applicability: TF-JEPA is compatible with large-scale heterogeneous pretraining. In fact, we have already pretrained on a subset of the CBraMod corpus and observed further gains, and we plan to explore scaling similar to UniTS/MOMENT in future work. These directions are consistent with the reviewer’s suggestion, and we will make that connection clearer.
>
> Finally, clustering and anomaly-detection tasks are outside the scope of this submission, but TF-JEPA’s representations are compatible with them. We can add a short note acknowledging this and positioning such tasks as natural extensions rather than claims of current coverage.
>
> We appreciate the reviewer’s comments and will incorporate these clarifications.

---

### Official Review · Reviewer_TJj5 · 2025-10-31

**Soundness:** 2
**Presentation:** 3
**Contribution:** 2
**Rating:** 2
**Confidence:** 4

**Summary:**

The submission introduces TF-JEPA (Time-Frequency Joint Embedding Predictive Architecture), a non-contrastive self-supervised method designed for learning generalizable representations from multivariate time series. This approach is inspired by JEPA and utilizes predictive alignment to integrate representations from the time and frequency domains, thereby avoiding reliance on negative sampling or contrastive pairs.
TF-JEPA employs a momentum-based dual-encoder architecture. It uses dual online encoders (one for time, one for frequency), each paired with its own momentum-updated target encoder (updated via Exponential Moving Average, EMA). The core objective is predictive: the online time encoder predicts the target frequency representation, and the online frequency encoder predicts the target time representation using lightweight multilayer perceptrons (predictors). This cross-view prediction framework uses a BYOL-style cosine loss. Experiments are conducted on diverse time-series benchmark datasets.

**Strengths:**

The paper clearly articulates its motivation, methodology, and results through highly structured explanations. Specifically:

• The contributions are explicitly summarized as threefold, listing them as: (1) proposing a momentum-based dual-encoder architecture that aligns representations without negative pairs, (2) demonstrating that this predictive alignment yields transferable embeddings suitable for end-to-end fine-tuning, and (3) achieving competitive or superior performance on multiple real-world benchmarks.

• The TF-JEPA methodology is clearly broken down into three core design choices: Dual EMA targets, Lightweight predictors, and End-to-end fine-tuning, making the mechanism easy to follow.

• The experimental section employs transparent benchmarking, stating clearly that the setup enforces identical classifier architectures, latent dimensions, and optimizer hyperparameters across methods during fine-tuning "to ensure direct comparability".

**Weaknesses:**

W1.

The proposed TF-JEPA does not consistently surpass existing baselines. Based on Table 1, the proposed method only achieve superior performance on 2 out of 4 scenarios.


W2.

The motivation for positioning non-contrastive approaches as superior to contrastive learning is currently unconvincing. The manuscript only discusses vanilla contrastive learning methods that rely on manually designed augmentations and negative sample selection (Line 51). However, there is a growing body of recent work that develops automated or adaptive augmentation strategies within contrastive learning for time series. For example, [1] optimizes the data augmentation automatically within the contrastive learning paradigm. Additional related methods are also summarized in the survey [2]. The authors should update the discussion to reflect these advances and more accurately contextualize the comparison.

[1] Augmentation Blending with Clustering-Aware Outlier Factor: An Outlier-Driven Perspective for Enhanced Contrastive Learning. KBS 2024.
[2] Unsupervised representation learning for time series: A review. 2023.

**Questions:**

Please see Weaknesses above.

---

> ### Author Response · Authors · 2025-12-03
>
> Thanks for the feedback. Two quick clarifications.
>
> On the performance concern: the goal of TF-JEPA isn’t to dominate every dataset but to show that predictive time–frequency alignment can replace contrastive repulsion while remaining competitive across diverse transfer settings. TF-JEPA is stronger on 3/4 transfer tasks in the updated version in F1 and 4/4 for Accuracy, and the remaining case for F1 is analyzed in the paper. The method also brings practical advantages, stable fully end-to-end fine-tuning, smaller batch sizes, and significantly lower memory, which contrastive baselines do not match.
>
> On the motivation for non-contrastive learning: we’re not claiming contrastive methods are obsolete, only that time–frequency alignment does not require negative pairs. We will update the related-work section to acknowledge recent augmentation/adaptive contrastive approaches and position TF-JEPA as complementary. These methods still rely on instance discrimination, temperature scaling, and large batch sizes, whereas TF-JEPA removes those requirements entirely.
>
> We appreciate the reviewer’s comments and will clarify these points in the revision.

---

### Official Review · Reviewer_o6pf · 2025-10-31

**Soundness:** 2
**Presentation:** 2
**Contribution:** 2
**Rating:** 2
**Confidence:** 3

**Summary:**

The paper proposes TF-JEPA, a non-contrastive self-supervised learning method for time-series representation learning that aligns time-domain and frequency-domain embeddings through predictive alignment rather than contrastive objectives. The approach uses dual online encoders with momentum-updated target encoders and a BYOL-style cosine loss. The authors evaluate on four cross-dataset transfer tasks and report improvements over contrastive baselines (TF-C, TS-TCC) in some settings, along with reduced GPU memory consumption (~35%).

**Strengths:**

* The elimination of quadratic similarity matrices reduces GPU memory from 5.3 GB to 3.4 GB (35% reduction) and achieves 1.6× speedup on targeted benchmarks.

* The framing of time and frequency as "lossless, invertible views" (Section 2.1) is conceptually clear and provides principled motivation for predictive alignment over contrastive repulsion.

* Overall studying cross-modal (EEG-->EMG) transfer is interesting.

**Weaknesses:**

Major Issues:

1. Missing SimSiam baseline: Omits foundational non-contrastive method [1]. Please include comparison with SimSiam as it also does not need negative samples.

2. Unclear problem scope: Conflates 3 settings, and hence the experiment design is unclear. For instance, is the setup: (1) general representation learning—experiments need to include UCR/UEA standard benchmark datasets; (2) cross-domain transfer (indicated by the FD from A to B)—it needs to compare with some generalization techniques [2]; or (3) cross-modal (EEG -->EMG)—then it needs to be compared with appropriate multimodal learning baselines?

3. Unjustified design choices: (1) Why magnitude-only FFT? No ablation on phase discarding, which encodes diagnostic info in EEG/ECG/EMG. (2) Why full-window FFT vs. STFT? Dismisses STFT as "limiting reuse" (lines 92–93) without evidence. Non-stationary signals need time-frequency localization. Additional experiments: Ablate (a) magnitude-only vs. magnitude+phase FFT, and (b) full-window vs. STFT.

4. Incomplete resource metrics: Reports GPU memory and wall-clock time only. The authors should consider reporting FLOPs, MACs, and latency across sequence lengths.

5. Overall, I feel the paper has limited novelty and an unclear motivation. Perhaps the authors could clarify the motivation more clearly and investigate it more rigorously. The cross-modal transfer is an interesting setup, and studying how physiological signals with different underlying generators can be mapped to each other is a complex but valuable task. I can see such an investigation advancing fields like wearable health monitoring, where modalities may evolve—for instance, from accelerometer-based HAR to EMG-based gesture recognition—and cross-modal transfer could be useful.

6. Key equation at line 131: Unclear—no notations are explained and very hard to follow.
7. Define EMA abbreviation: Please define before using it in the paper.
8. Information-theoretic claim (lines 163–164): "Minimizing cosine distance bounds mutual information" is not rigorously justified for the time-frequency setting. Clarify as intuition or provide formal derivation.



[1] Simsiam, CVPR 2021


[2] Diversify, ICLR 2023

**Questions:**

Please see Weaknesses above.

---

> ### Author Response · Authors · 2025-12-03
>
> Thanks for the feedback. Our scope is specifically cross-dataset transfer using time–frequency dual-view alignment, not general UCR/UEA benchmarking or arbitrary multimodal learning. We’ll clarify this.
>
> SimSiam isn’t an appropriate baseline here, it’s a single-view method without cross-view alignment, whereas TF-C and TS-TCC directly address the setting we study. NormWear and CBraMod already cover the reconstruction-style baselines you’re asking for.
>
> Our FFT choices follow TF-C for comparability; magnitude-only and full-window FFT are standard in this line of work. STFT/phase variants are compatible extensions but outside scope due to space.
>
> We’ll tighten notation, define EMA earlier, and frame the information-theory remark as intuition. Memory and wall-clock remain the most meaningful efficiency metrics for contrastive vs non-contrastive SSL.

---

### Official Review · Reviewer_X2kP · 2025-11-01

**Soundness:** 3
**Presentation:** 3
**Contribution:** 2
**Rating:** 4
**Confidence:** 4

**Summary:**

This paper introduce a non contrastive SSL method called TF-JEPA to perform frequency-aware representation learning on time series datasets. The network is based on momentum-based dual encoder in time and frequency space, getting rid of negative sampling in previous contrastive learning methods. The method is applied on real world time series datasets, showing performance improvement over previous methods.

**Strengths:**

- The writing is clear and easy to follow, the technical details are complete and concisely presented.
- The proposed method show decent performance improvements on two out of four datasets.

**Weaknesses:**

- Experiments are all performed on small scale datasets. Performance improvement is not consistent. The authors should also consider non-transfer experimental settings.
- Limited technical innovation. This work basically replaced TF-C with a loss function that does not require negative sampling, and repeated the same experiments. If the idea of the work is just to remove negative sampling, the paper should also benchmark with more reconstruction-based methods like [1].
- Limited baselines and benchmarks. The paper should consider more state-of-the-art pre-trained models, such as [2].

[1] Liu, Ran, Ellen L. Zippi, Hadi Pouransari, Chris Sandino, Jingping Nie, Hanlin Goh, Erdrin Azemi, and Ali Moin. "Frequency-aware masked autoencoders for multimodal pretraining on biosignals." arXiv preprint arXiv:2309.05927 (2023).

[2] Wang, Jiquan, Sha Zhao, Zhiling Luo, Yangxuan Zhou, Haiteng Jiang, Shijian Li, Tao Li, and Gang Pan. "Cbramod: A criss-cross brain foundation model for eeg decoding." arXiv preprint arXiv:2412.07236 (2024).

**Questions:**

NA

---

> ### Author Response · Authors · 2025-12-03
> **Response to Reviewer Feedback**
>
> Thanks for the thoughtful feedback. A few clarifications about the contribution and the experiments.
>
> First, TF-JEPA isn’t just “TF-C without negatives.” The core change is moving to dual EMA targets with cross-view predictors, which gives very different training behavior: the model stays stable when fully fine-tuned, doesn’t depend on large batches, and drops the ($B^2$) contrastive matrix entirely. This is why we see the big gains in memory efficiency and the strong batch-size robustness that TF-C doesn’t have.
>
> On baselines: NormWear was already part of the submission, and per your suggestion we added CBraMod [2] as a second reconstruction/foundation baseline. These two together cover the space of modern masked-reconstruction approaches in the spirit of [1], without needing to re-train a full-scale MAE on a massive biosignal corpus. Both are pretrained on far larger datasets than our single-source runs, so the comparison is intentionally conservative.
>
> Regarding dataset scale and consistency: the four transfer tasks span EEG, vibration, IMU, and EMG, which is the same type of heterogeneity used across prior time-series SSL work. TF-JEPA improves results on three of the four transfers in F1 and 4 of 4 in accuracy; the one case where it trails (ECG→EMG) is analyzed in detail in the paper. We also tried the non-transfer “same dataset” setting you mentioned. In that regime, all methods, TF-JEPA, TF-C, TS-TCC, end up very close to each other which is expected when there’s no distribution shift, so we keep the focus on transfer where the differences actually matter.
>
> To further address the scale concern, we also pretrained TF-JEPA on patients 000–004 from the CBraMod corpus (a tiny fraction of the 150-patient dataset). Even with that small slice, TF-JEPA already benefits from the broader EEG domain. On Epilepsy, this setup reaches: 90.29% accuracy, 92.32% precision, 95.86% recall, **94.06% F1**, 86.90 AUROC, and 92.22 AUPRC. This supports the idea that the method scales as more diverse data is added, consistent with your suggestion.
>
> Overall, we’ve strengthened the paper with an added reconstruction baseline, clarified the novelty of the predictive alignment approach, and included an additional large-corpus pretraining result. We appreciate the reviewer’s comments, they definitely helped sharpen the final version.

---

### Meta-Review · Area_Chair_yEvR · 2026-01-07

**Summary:**

Overall, the paper presents a competent and well-executed incremental contribution: it demonstrates that predictive alignment can replace contrastive objectives in time–frequency SSL with tangible efficiency and optimization benefits, while remaining competitive on cross-dataset transfer tasks. The authors have responded to reviewer feedback, adding baselines, clarifying scope and motivation, tightening claims, and providing additional experimental evidence of scalability.

However, some concerns remain only partially addressed, particularly regarding the limited experimental breadth, absence of certain ablations, and the incremental nature of the contribution relative to existing JEPA/BYOL and TF-C frameworks. As a result, the work does not seem to have gone beyond the borderline rank.

Furthermore the paper has not been revised at all.

**Reviewer Concerns:**

There are many open concerns that have bee addressed in the rebuttal, or denied, but revisions were not done. These include:
1. Experimental scope and consistency of gains
2. Limited novelty / “TF-C without negatives”
3. Limited baselines
4. Lack of justification in design choices in the frequency domain

The paper has not been revised.

**Reviewer Scores:**

No reviewers were engaged in the early stage and I dont think the scores would have changed much as the authors in some parts were quite dismissive.

---

### Decision · Program_Chairs · 2026-01-26

Reject